# Multi-task Graph Neural Architecture Search with Task-aware Collaboration and Curriculum

**Yijian Qin**[1,2], **Xin Wang**[1,2]*, **Ziwei Zhang**[1], **Hong Chen**[1], **Wenwu Zhu**[1,2]*

[1]Department of Computer Science and Technology, Tsinghua University, [2]BNRist[†]

qinyj19@mails.tsinghua.edu.cn, {xin_wang,zwzhang}@tsinghua.edu.cn
h-chen20@mails.tsinghua.edu.cn, wwzhu@tsinghua.edu.cn

## Abstract

Graph neural architecture search (GraphNAS) has shown great potential for automatically designing graph neural architectures for graph related tasks. However, multi-task GraphNAS, capable of handling multiple tasks simultaneously and capturing the complex relationships and dependencies between them, has been largely unexplored in literature. To tackle this problem, we propose a novel multi-task graph neural architecture search with task-aware collaboration and curriculum (MTGC$^3$), which is able to simultaneously discover optimal architectures for different tasks and learn the collaborative relationships among different tasks in a joint manner. Specifically, we design the structurally diverse supernet to manage multiple architectures and graph structures in a unified framework, which combines with our proposed soft task-collaborative module to learn the transferability relationships between tasks. To further improve the architecture search procedure, we develop the task-wise curriculum training strategy that reweighs the influence of different tasks based on their relative difficulties. Extensive experiments show that our proposed MTGC$^3$ model achieves state-of-the-art performance against several baselines in multi-task scenarios, demonstrating its ability to discover effective architectures and capture collaborative relationships for multiple tasks.

## 1 Introduction

Graph-structured data has attracted lots of attentions in recent years for its flexible representation ability in various domains. Graph neural networks (GNNs) such as GCN [17], GAT [39], and GIN [47] have been proposed and achieved great success in many graph applications. To save human efforts on designing GNN architectures for different tasks, graph neural architecture search (GraphNAS) [19, 7, 42] has been proposed to search for optimal GNN architectures. These automatically designed GNN architectures have achieved competitive or better performances compared with manually designed GNNs for a single task.

On the other hand, multi-task scenarios are ubiquitous for graph data. For example, in drug discovery, predicting multiple properties of molecules can be regarded as multiple tasks. Properly capturing the relationship between these tasks allows for better learning of them. Nevertheless, there has been no work reported on automatically searching GNN architectures for multi-task scenarios in the literature.

In this paper, we study the important yet unexplored problem of multi-task GraphNAS for the first time. Given that different GNNs employ different architectures for adaption to various graph tasks, we tackle this problem through jointly customizing an architecture for each individual task and considering the task collaborations to share useful information among different tasks. However, discovering

---

*Corresponding Authors.

[†]BNRist is the abbreviation of Beijing National Research Center for Information Science and Technology.

37th Conference on Neural Information Processing Systems (NeurIPS 2023).

the optimal GNN architectures for multiple tasks poses several challenges: (1) It is non-trivial to manage multiple architectures within a supernet, which is crucial to guarantee the effectiveness of the neural architecture search; (2) It is challenging to capture the complex relationships among different tasks for information sharing during the searching process; (3) It remains a challenge to balance the influence of different tasks on architecture optimization since different tasks may show diverse difficulties and optimization patterns.

To address these challenges, we propose the multi-task graph neural architecture search with task-aware collaboration and curriculum (MTGC[3])[3] in this paper. Our proposed MTGC[3] model is able to discover multiple optimal architectures within a unified supernet and simultaneously optimizes them through learning the collaborative relationships of different tasks in a joint manner. Specifically, we design the *structurally diverse supernet* to manage multiple architectures and graph structures within a supernet for effective architecture search. Then, we introduce the *soft task-collaborative module* to capture the complex relationships among tasks, enabling sharing useful information during the searching process. We further develop the *task-wise curriculum training strategy* to better optimize the supernet via reweighing the influence of different tasks on architecture optimization based on task difficulties. Extensive experiments on both synthetic and real-world datasets validate the superiority of our proposed MTGC[3] model over existing baselines via customizing the optimal architecture for each task and sharing useful information among them. Detailed ablation studies further verify the effective designs of MTGC[3].

Our contributions are summarized as follows.

- We are the first to investigate the problem of multi-task graph neural architecture search via proposing the multi-task graph neural architecture search with task-aware collaboration and curriculum (MTGC[3]), to the best of our knowledge.
- We propose to jointly discover optimal architectures for multiple tasks and capture their complex relations within a unified framework by designing i) the structurally diverse supernet, ii) the soft task-collaborative module, and iii) the task-wise curriculum training strategy.
- Extensive experimental results demonstrate that our proposed MTGC[3] model outperforms state-of-the-art baselines on both synthetic and real-world datasets.

## 2   Problem Formulation and Preliminaries

### 2.1   Message-passing graph neural network

GNNs are state-of-the-art models for graph machine learning, which typically follow a message passing scheme where nodes aggregate information from their neighbors in each layer formulated as:

$$\mathbf{m}_i^{(l)} = \text{Agg}(\mathbf{h}_j^{(l)} | j \in \mathcal{N}_i) \tag{1}$$

$$\mathbf{h}_i^{(l+1)} = \text{Update}(\mathbf{m}_i^{(l)}), \tag{2}$$

where $\mathbf{h}_i^{(l)}$ is the representation of node $i$ at the $l$-th layer, $\mathcal{N}_i$ denotes the neighbors of node $i$ derived from the adjacent matrix, $\text{Agg}(\cdot)$ is the aggregation function, $\text{Update}(\cdot)$ is an updating function between two node representations. Different GNNs have different aggregation and updating functions, which is the main search objective for GraphNAS.

### 2.2   Multi-task graph neural architecture search

Given a set of graph tasks $\{T_m\}_{m=1}^M$, where each task has a set of graphs $G_m = \{g_m^1, g_m^2, \cdots, g_m^{n_m}\}$ in a shared graph space $\mathcal{G}$ and the labels $Y_m = \{y_m^1, y_m^2, \cdots, y_m^{n_m}\}$ in its own label space $\mathcal{Y}_m$. We denote $\mathcal{Y}$ as the composition of all label spaces, i.e., $\mathcal{Y} = \bigcup \{\mathcal{Y}_m\}_{m=1}^M$. The goal of multi-task graph learning is to design a model $F : \mathcal{G} \to \mathcal{Y}$ to map the graphs to the label space while minimizing the loss function, i.e.,

$$\arg\min_F \mathcal{L}(\{F(G_m), Y_m\}_{m=1}^M), \tag{3}$$

where $\mathcal{L}$ is the multi-task loss function. Here we use multi-task graph classification as an example, but the definition can be easily extended to multi-task node classification problems. In GraphNAS,

---

[3]`https://github.com/THUMNLab/AutoGL-light`

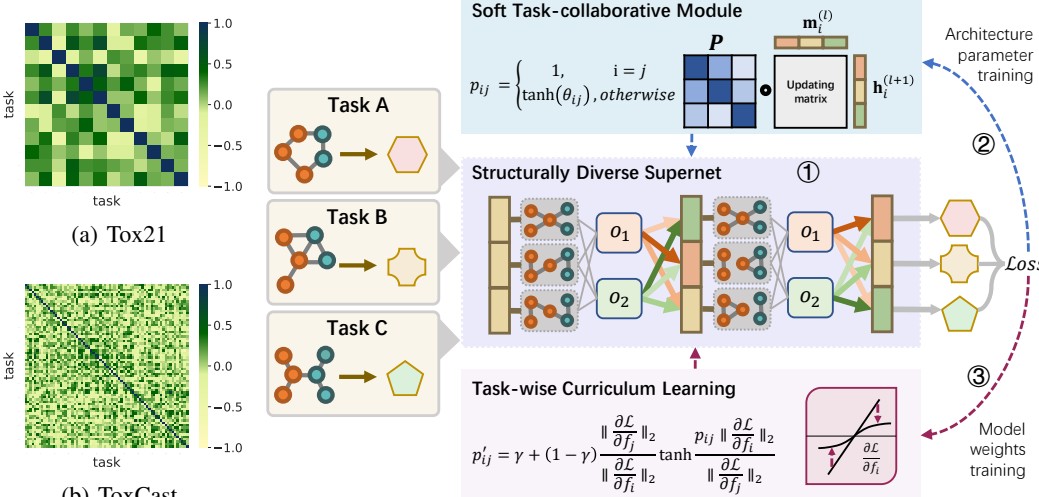

Figure 1: Kendall correlation coefficient of the performance ranks of different architectures on different tasks.

(a) Tox21

(b) ToxCast

Figure 2: An overview of our proposed MTGC$^3$ method. **Step 1:** Performing forward propagation through the structurally diverse supernet and the soft task-collaborative module. **Step 2:** Updating architecture parameters and the soft task-collaborative module. **Step 3:** Updating model weights through the task-wise curriculum learning.

we mainly focus on the function $f$ being GNNs. A typical GNN consists of two parts: an architecture $\alpha \in \mathcal{A}$ and learnable weights $w \in \mathcal{W}$, where $\mathcal{A}$ and $\mathcal{W}$ denotes the architecture space and the weight space, respectively. Therefore, we denote GNNs in Eq.(1) as the following mapping function: $F_{\alpha,w} : \mathcal{G} \to \mathcal{Y}$. In this paper, we focus on the aggregation functions $\text{Agg}(\cdot)$. Specifically, we consider a search space of standard layer-by-layer architectures without sophisticated connections such as residual or jumping connections, though our proposed method can be easily generalized. We choose five widely used message-passing GNN layers as our operation candidate set $\mathcal{O}$, including GCN [17], GAT [39], GIN [47], SAGE [11], k-GNN [27], and ARMA [12]. Besides, we also adopt MLP, which does not consider graph structures.

## 3 The Proposed Method

To develop a GraphNAS method for multi-task learning, we first consider what designs should be included that benefit multi-task learning. We propose an assumption on which our method is based.

**Assumption 3.1.** Different GNN architectures are suitable for learning different graph tasks. For example, if we search architectures separately for different tasks, i.e., the following bi-level optimization problem:

$$
\begin{aligned}
\alpha_m &= \arg \min_{\alpha} \mathcal{L}(F_{\alpha, w(\alpha)}(G_m), Y_m) \\
w(\alpha) &= \arg \min_{w} \mathcal{L}(F_{\alpha, w}(G_m), Y_m),
\end{aligned}
\tag{4}
$$

where $\alpha_m$ is the most suitable architecture for task $T_m$. Then, the optimal architecture $\alpha_m$ can be different for different tasks.

Assumption 3.1 implies that different tasks have different characteristics, while these characteristics require different architectures. To validate this assumption, we randomly choose several architectures and evaluate them on different tasks individually on two graph datasets: Tox21 and ToxCast (refer to Section 4.1 for details). We rank these architectures according to their performance for each task. Then we use Kendall rank correlation to compute the similarity score between rankings of the architectures for all pairs of tasks. The result is shown in Figure 1. We can find that many tasks have a negative correlation between different architectures, indicating that different architectures behave very differently on different tasks. The result illustrates the necessity to customize an architecture for different tasks instead of using a shared architecture.

On the other hand, it is also important to share useful information between different tasks, which is a common and basic assumption in multi-task learning [3]. Combining these two aspects, our goal is to simultaneously search multiple architectures for different tasks while learning the collaborative relationships of them. Figure 2 shows the overall framework of our method. In the following subsections, we present our proposed method in detail.

## 3.1 Structurally diverse supernet

Based on Assumption 3.1, we need to search multiple architectures for different tasks simultaneously. To effectively manage multiple architectures, we propose the structurally diverse supernet, following the widely adopted supernets in NAS [29]. The design principle is that, although different tasks require different architectures, we cannot completely separate these architectures since they need to share useful knowledge. We propose to balance this tradeoff by disentangling the supernet. In our structurally diverse supernet, the node representation at each layer is divided into $N$ chunks with different GNN operations. In addition, we make the graph structure to be diverse in different chunks to further disentangle the hidden features and help different chunks to learn more specific knowledge for the corresponding tasks. A disentangled layer can be formulated as follows:

$$f(\mathbf{x}, \mathbf{A}) = \|_{i=1}^{N} f_i(\mathbf{x}, \mathbf{A}_i), \tag{5}$$

where $f$ one of the layers in the mapping function $F$, $\mathbf{x}$ is the hidden features given by the last layer, and $\mathbf{A}$ is the adjacent matrix. Moreover, $\|$ represents concatenation, $\mathbf{A_i}$ is the adjacent matrix of the graph used in the $i$-th chunk, $f_i$ denotes the GNN operation of the $i$-th chunk. By stacking the disentangled layers into a GNN, we can capture a variety of latent factors of the graphs by different types of GNN operations and graph structures, which is beneficial for multi-task learning.

Following the NAS literature [24], we adopt supernets to search for the optimal architecture in a differentiable way. Specifically, in the supernet, all possible operations are jointly considered by mixing different operations into a mixed operation, i.e., $f(\mathbf{x}) = \sum_{o_k \in \mathcal{O}} \alpha_k o_k(\mathbf{x})$, where $o_k$ is a candidate operation and $\mathcal{O}$ is the operation search space, $\alpha$ denotes the architecture parameters to be optimized. We combine the DARTS manner with our structurally diverse supernet to enable differentiable optimization, which can be formulated as follows:

$$f_i(\mathbf{x}, \mathbf{A}_i) = \sum_{o_k \in \mathcal{O}} \alpha_{ik} o_{ik}(\mathbf{x}, \mathbf{A}_i), \tag{6}$$

where $\alpha_{ik}$ is the architecture parameter of the $k$-th operation at the $i$-th chunk. As a result, multiple architectures in different chunks are contained in the supernet and they can also transfer knowledge to each other.

To adopt diverse structures for different chunks, we use a parameter-efficient and differentiable manner to generate the weights of edges for the graph structures, which can be formulated as follows:

$$A_{i,(u,v)} = \delta A_{(u,v)} + (1 - \delta) A_{(u,v)} \cdot \text{sigmoid}((\mathbf{S}_i^u \mathbf{x}_u)^\top (\mathbf{S}_i^v \mathbf{x}_v)), \tag{7}$$

where $A_{i,(u,v)}$ is the weight of edge between node $u$ and $v$ in $\mathbf{A}_i$, $A_{(u,v)}$ indicates if there is an edge between node $u$ and $v$ in the original graph, $\mathbf{S}_i^u$ and $\mathbf{S}_i^v$ are learnable parameters to generate edge weights of the $i$-th chunk, $\delta$ is the hyper-parameter. Consequently, we generate an adjacent matrix $\mathbf{A}_i$ in a continuous space. We can conveniently use gradient based methods to optimize it.

## 3.2 Soft task-collaborative module

Using the structurally diverse supernet, different chunks can exchange information to share knowledge. However, the independence of chunks is not encouraged, and therefore representations could be potentially mixed together. To further ensure that different chunks of the supernet correspond to relatively independent architectures, we propose a soft task-collaborative module. First, we rewrite Eq. (6) as follows:

$$f_i(\mathbf{x}, \mathbf{A}_i) = \sum_{o_k \in \mathcal{O}} \alpha_{ik} \sum_j o_{ijk}(\mathbf{x}_j, \mathbf{A}_i), \tag{8}$$

where $\mathbf{x}_j$ is the output of the $j$-th chunk of the previous layer, and $o_{ijk}$ is the GNN operation only containing the corresponding parameters. For example, many types of GNN operations in Eq. (6) contain a parameter matrix with the shape $d \times d_c$ in the node updating function in Eq. (2), where

$d_c$ is the dimensionality of each chunk and $d = N \times d_c$ is the overall dimensionality. For these operations, we can split the parameter matrix into $N$ matrices with the shape $d_c \times d_c$, where the $j$-th matrix corresponds to the part of $\mathbf{x}_j$. We denote $o_{ijk}$ as the GNN operation only containing this part of parameters.

Using Eq.(8), we introduce our proposed soft task-collaborative module. Our goal is to keep the independence of each chunk, i.e., the $i$-th chunk of node representation at the previous layer mainly contributes to the $i$-th chunk of node representation at the next layer. We formulate it as follows:

$$f_i(\mathbf{x}, \mathbf{A}_i) = \sum_{o_k \in \mathcal{O}} \alpha_{ik} \sum_j p_{ij} o_{ijk}(\mathbf{x}_j, \mathbf{A}_i), \quad p_{ij} = \begin{cases} 1 & \text{if } i = j, \\ \tanh \theta_{ij} & \text{otherwise,} \end{cases} \tag{9}$$

where $\theta$ is learnable parameters. $p_{ij}$ represents the impact of the $j$-th chunk of node representation at the previous layer to the $i$-th chunk of node representation at the next layer. We remark that the absolute value of $\tanh \theta$ indicates the amount of contribution while the sign of $\tanh \theta$ indicates the positivity or negativity of the correlation. Since $\tanh \theta \in (-1, 1)$, we ensure that node representation from the same chunk has the largest impact on the next layer. Therefore, the architectures of different chunks are relatively independent, which helps to make the hidden representation disentangled and contain more useful information for different tasks. Besides, we initialize $\theta$ as $0$ so that the initial structurally diverse supernet simply consists of several separated GNN architectures at the beginning. By updating $\theta$ in the training procedure, our proposed method can automatically learn the transferability between architectures.

After learning representations containing diverse information by the supernet, our model makes predictions for multiple tasks. We propose two approaches to building the task head.

**Task-separated head**: We manually pair each chunk of the supernet to a task, i.e., the node representation of the last layer of the supernet is split according to the chunks, and each chunk is connected to the head of the corresponding task. In this case, the architecture of each chunk can be regarded as the customized architecture for the corresponding task, and $p_{ij}$ represents the collaborative pattern between different tasks, i.e., whether to share useful information between them. Since the number of chunks needs to be equal to the number of tasks, this approach is only applicable when the number of tasks is not too large.

**Cross-mixed head**: When the number of tasks is large, using the task-separated head will result in the dimensionality of the representation growing too large. Therefore, we further propose the cross-mixed head by randomly pre-assigning a few dimensions to each task, i.e., the head of each task is only connected to the representations of these dimensions. In practice, we overlay a mask tensor sampled from a Bernoulli distribution on the weights of the classification heads. This approach allows each task to acquire different hidden features while preventing some tasks from dominating all the architectures in the supernet.

### 3.3 Task-wise curriculum training

Our framework is end-to-end and can be optimized by the multi-task loss. But using multi-task loss may cause difficulties in training due to task imbalances [5]. At first we give the proposition below to analyze the gradient of different chunks in our model. We denote $w_{j'}$ as the parameters that generate $\mathbf{x}_j$ in the last layer.

**Proposition 3.2.** *When using the overall loss function of multiple tasks $\mathcal{L}$ for gradient back-propagation in our framework, the partial derivative of $\mathcal{L}$ with respect to $w_{j'}$ is:*

$$\frac{\partial \mathcal{L}}{\partial w_{j'}} = \sum_i p_{ij'} \frac{\partial \mathcal{L}}{\partial f_i} \sum_{o_k \in \mathcal{O}} \alpha_{ik} \frac{\partial o_{ij'k}(\mathbf{x}_{j'}, \mathbf{A}_i)}{\partial w_{j'}}. \tag{10}$$

For the proof of Proposition 3.2, please refer to Appendix B. In Eq. (10), we can regard the latter part of the right side $\sum_{o_k \in \mathcal{O}} \alpha_{ik} \frac{\partial o_{ij'k}(\mathbf{x}_{j'}, \mathbf{A}_i)}{\partial w_{j'}}$ as the partial derivative of the mixed operation in a classical supernet in Eq. (6), and $\frac{\partial \mathcal{L}}{\partial w_{j'}}$ is the sum of these partial derivatives weighted by a score $p_{ij'} \frac{\partial \mathcal{L}}{\partial f_i}$. However, different tasks may have different difficulties and show diverse patterns in the optimization process, e.g., tasks are unbalanced in terms of backpropagated gradient scales, some tasks with large score $p_{ij'} \frac{\partial \mathcal{L}}{\partial f_i}$ may dominate the training process. Besides, in our model, the optimization

---

**Algorithm 1:** MTGC$^3$

---

**Input:** Training dataset $\{G_i\}_{i=1}^M$ and $\{Y_i\}_{i=1}^M$, learning rates $\eta_\alpha$, $\eta_\mathbf{S}$, $\eta_\theta$ and $\eta_w$

1   Initialize learnable parameters $w$, $\alpha$, $\mathbf{S}$ and $\theta$. Set $\gamma = 0$;

2   **while** *not converge* **do**

3      Forward propagate by Eq.(5) and Eq.(9) to calculate loss $\mathcal{L}$;

4      $\alpha \leftarrow \alpha - \eta_\alpha \nabla_\alpha \mathcal{L}(w, \alpha, \mathbf{S}, \theta)$,   $\mathbf{S} \leftarrow \mathbf{S} - \eta_\mathbf{S} \nabla_\mathbf{S} \mathcal{L}(w, \alpha, \mathbf{S}, \theta)$,   $\theta \leftarrow \theta - \eta_\theta \nabla_\theta \mathcal{L}(w, \alpha, \mathbf{S}, \theta)$;

5      Using $\frac{\partial \mathcal{L}}{\partial f_j}$ to calculate $p'$ by Eq.(11);

6      Forward propagate by Eq.(5) and Eq.(11) to calculate loss $\mathcal{L}'$;

7      $w \leftarrow w - \eta_w \nabla_w \mathcal{L}'(w, \alpha, \mathbf{S}, \theta)$;

8      Increase $\gamma$;

9   **end**

---

curves of different chunks can also differ greatly, which makes the optimization of our model further challenging.

To tackle these problems, we propose the task-wise curriculum training strategy. Specifically, we modify Eq. (9) using a curriculum strategy as:

$$f_i'(\mathbf{x}, \mathbf{A}_i) = \sum_{o_k \in \mathcal{O}} \alpha_{ik} \sum_j p_{ij}' o_{ijk}(\mathbf{x}_j, \mathbf{A}_i), \quad p_{ij}' = \gamma p_{ij} + (1 - \gamma) \frac{\| \frac{\partial \mathcal{L}}{\partial f_j} \|_2}{\| \frac{\partial \mathcal{L}}{\partial f_i} \|_2} \tanh \frac{p_{ij} \| \frac{\partial \mathcal{L}}{\partial f_i} \|_2}{\| \frac{\partial \mathcal{L}}{\partial f_j} \|_2}, \tag{11}$$

where $\gamma$ is a parameter that increases from 0 to 1 during the training process, and $\| * \|_2$ is the $l_2$-normalization of the target partial derivative for measuring its scale. We use Eq. (11) instead of Eq. (9) while calculating the gradients of $w$. Using this curriculum strategy benefits the optimization process of our framework. When $\gamma = 0$, we have $p_{ij}' < \frac{\| \frac{\partial \mathcal{L}}{\partial f_j} \|_2}{\| \frac{\partial \mathcal{L}}{\partial f_i} \|_2}$, thus the contribution of tasks with large partial derivatives to the loss function is suppressed. For tasks with small $\| \frac{\partial \mathcal{L}}{\partial f_j} \|_2$, the $\tanh$ function tends to be a linear function, thus $p_{ij}' \approx p_{ij}$ and their contributions are not affected. As $\gamma$ increases, Eq. (11) gradually converges to Eq. (9), keeping Eq. (11) an unbiased estimate of the original loss function. Note that gradients are often used in curriculum learning to measure the difficulty of data [40, 4, 55, 41, 53], but our method introduce this idea to measure the difficulty of tasks by gradients to implement task-aware curriculum learning and adjust the training process accordingly.

The overall training procedure is shown in Algorithm 1. The entire model can be learned end-to-end by gradient based approaches. We update the learnable parameters $w$, $\alpha$, $\theta$, and $\mathbf{S}$ in an iterative way. After training, we keep the architecture and parameters in the supernet for evaluation without the architecture discretization step, enhancing flexibility on architecture search and simplifying the optimization strategy.

## 4   Experiments

### 4.1   Experiment Setting

**Datasets**. We adopt both synthetic and real-world multi-task graph datasets.

- **Synthetic Datasets.** We construct a synthetic multi-task node classification dataset named Multi-GNN Label (MGL). We generate several MGL graphs using different graph generation methods, e.g., the Erdös-Rényi model (MGL-ER), the Watt-Strogatz small-world model (MGL-WS), and the Barabási-Albert preferential attachment (MGL-BA). The node features are sampled from a Gaussian distribution. To generate labels with diverse characteristics, we adopt different GNNs with random parameters to generate multiple hidden node representations. For each hidden representation, we use the KNN algorithm to group nodes into 5 classes. As such, each GNN can generate node labels corresponding to a node classification task. Specifically, we use two random MLP, GCN, GAT, and SAGE to generate 8 labels.

Table 1: The test accuracy of all the methods on the synthetic dataset MGL and the test ROC-AUC of all the methods on the real-world datasets OGBG. We run all experiments 10 times with different random seeds and report the average results with standard deviations. The best results are in bold.

| Dataset | MGL-ER | MGL-WS | MGL-BA | Tox21 | ToxCast | Sider |
|---|---|---|---|---|---|---|
| GCN | $51.50_{\pm0.37}$ | $50.17_{\pm0.14}$ | $50.41_{\pm0.21}$ | $76.41_{\pm0.60}$ | $64.91_{\pm0.53}$ | $62.02_{\pm1.71}$ |
| GAT | $50.95_{\pm0.17}$ | $49.91_{\pm0.19}$ | $47.77_{\pm0.52}$ | $75.62_{\pm0.89}$ | $63.52_{\pm0.57}$ | $58.57_{\pm1.98}$ |
| GIN | $52.40_{\pm0.41}$ | $47.67_{\pm0.28}$ | $50.48_{\pm0.45}$ | $77.20_{\pm0.85}$ | $64.58_{\pm0.49}$ | $58.48_{\pm1.63}$ |
| SAGE | $62.58_{\pm0.13}$ | $64.77_{\pm0.14}$ | $66.15_{\pm0.25}$ | $75.87_{\pm0.49}$ | $64.08_{\pm0.65}$ | $60.12_{\pm0.97}$ |
| k-GNN | $56.49_{\pm0.29}$ | $58.80_{\pm0.14}$ | $62.19_{\pm0.47}$ | $76.87_{\pm0.51}$ | $63.86_{\pm0.59}$ | $60.07_{\pm1.14}$ |
| ARMA | $63.90_{\pm0.29}$ | $64.20_{\pm0.28}$ | $65.05_{\pm0.37}$ | $75.93_{\pm0.87}$ | $64.36_{\pm0.83}$ | $60.79_{\pm1.24}$ |
| MLP | $50.08_{\pm0.13}$ | $40.93_{\pm0.13}$ | $43.63_{\pm0.20}$ | $74.47_{\pm0.75}$ | $62.67_{\pm0.61}$ | $60.70_{\pm1.29}$ |
| Random | $57.40_{\pm0.22}$ | $60.80_{\pm0.19}$ | $59.15_{\pm0.17}$ | $76.20_{\pm0.26}$ | $64.89_{\pm0.60}$ | $58.87_{\pm1.38}$ |
| DARTS | $64.94_{\pm0.20}$ | $64.16_{\pm0.29}$ | $62.39_{\pm0.23}$ | $76.96_{\pm0.57}$ | $65.23_{\pm0.60}$ | $60.64_{\pm1.37}$ |
| GNAS | $49.76_{\pm2.95}$ | $61.17_{\pm0.00}$ | $64.28_{\pm0.95}$ | $74.97_{\pm0.41}$ | $61.85_{\pm1.07}$ | $57.11_{\pm1.31}$ |
| PAS | - | - | - | $75.45_{\pm0.47}$ | $63.85_{\pm0.35}$ | $59.31_{\pm1.48}$ |
| GRACES | - | - | - | $74.82_{\pm0.85}$ | $65.77_{\pm0.53}$ | $61.85_{\pm2.56}$ |
| MTGC[3] | $\mathbf{66.33_{\pm0.34}}$ | $\mathbf{67.39_{\pm0.42}}$ | $\mathbf{68.36_{\pm0.22}}$ | $\mathbf{77.99_{\pm0.42}}$ | $\mathbf{66.36_{\pm0.26}}$ | $\mathbf{62.08_{\pm1.76}}$ |

- **Real-world Datasets.** We choose three widely-used multi-task graph classification datasets included OGB [13]: OGBG-Tox21 [14], OGBG-ToxCast [35], and OGBG-Sider [18]. These datasets contain a set of properties of toxicological assays or adverse drug reactions of drug molecules, where molecules can be represented as graphs and each task corresponds to predicting a property of drug molecules.

**Baselines**. We compare our model with 12 baselines from the following three different categories.

- **Manually Design GNNs**: we include the GNNs in our search space (refer to Section 2.2) as our baselines, i.e., GCN, GAT, GIN, SAGE, k-GNN, and ARMA. We also include MLP as a baseline.
- **Graph Neural Architecture Search**: We consider three representative GraphNAS baselines: GNAS [7], an reinforcement learning based method; PAS [42] and GRACES [32], two recent differentiable GraphNAS methods. Note that since GNAS and GRACES are specifically designed for graph classification tasks. We also consider two classic NAS baselines, random search and DARTS [24] combined with our search space.

**Experimental Details.** We set the number of layers as 3 for synthetic datasets, and 5 for real-world datasets. For all datasets except ToxCast, we use the task-separate head. For ToxCast, we use the cross-mixed head with 16 chunks.

**Number of Learnable Parameters.** Denote $|V|$, $|E|$ as the number of nodes and edges in the graph, respectively, and $d$ as the dimensionality of hidden representations. We denote $d_S$ as the dimensionality of the structure generation part, i.e., $\mathbf{S}$ is a matrix with the shape $d_S \times d$. The number of learnable parameters of a typical message-passing GNN is $O(d^2)$. In our framework, $w$, $\alpha$, $\theta$, and $\mathbf{S}$ has $O(|\mathcal{O}|d^2)$, $O(N|\mathcal{O}|)$, $O(d^2)$, and $O(Nd_sd)$ parameters, respectively. The total number of learnable parameters is $O(N(|\mathcal{O}| + d_Sd) + |\mathcal{O}|d^2)$. To guarantee fair comparison, we use small $d_S$ and $d$ in the experiments so that the different models are comparable in terms of the number of learnable parameters.

## 4.2 Qualitative Results

We summarize the experimental results in Table 1. For the results on synthetic datasets, our model outperforms all baselines in all three settings. Specifically, we find that most GNNs perform poorly, indicating that they cannot well handle multiple tasks. The existing GraphNAS methods fail to outperform some manually designed GNNs on these graphs. In contrast, MTGC[3] shows much better results by searching multiple architectures and capturing collaborative patterns between tasks.

As for the results on the three real-world multi-label graph classification benchmarks, we find that some NAS methods, e.g., DARTS and GRACES, achieve slightly better results than manually designed GNNs in some cases, demonstrating the importance of automating architectures. Nevertheless,

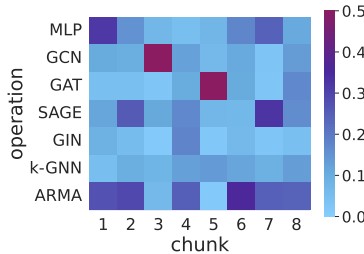

Figure 3: Visualizations of architecture parameters of different chunks in MGL-ER.

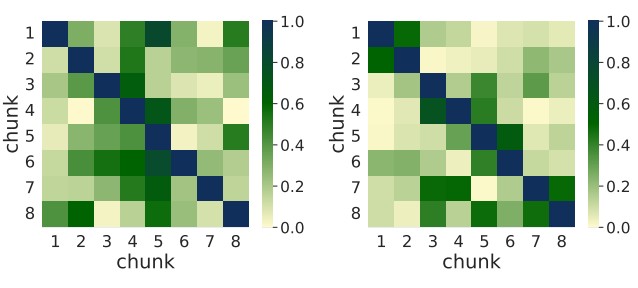

(a) The 1st layer        (b) The last layer

Figure 4: Visualizations of the absolute values of $p_{ij}$ of different layers in MGL-ER.

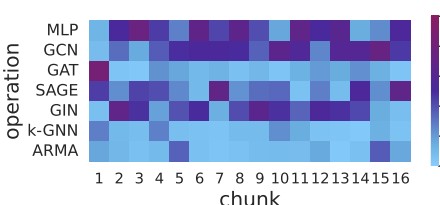

Figure 5: Visualizations of architecture parameters of different chunks at the 3rd layer in ToxCast.

| Variant | MGL-WS | Tox21 | ToxCast |
|---|---|---|---|
| NoStru | $67.17_{\pm 0.60}$ | $77.42_{\pm 1.00}$ | $66.30_{\pm 0.30}$ |
| Separate | $65.35_{\pm 0.86}$ | $76.20_{\pm 0.98}$ | $66.12_{\pm 0.80}$ |
| FullCollab | $67.21_{\pm 0.55}$ | $77.83_{\pm 0.71}$ | $66.01_{\pm 0.50}$ |
| MLPHead | $64.47_{\pm 1.16}$ | $77.62_{\pm 0.57}$ | $64.61_{\pm 0.53}$ |
| NoCL | $67.15_{\pm 0.52}$ | $77.68_{\pm 0.48}$ | $66.00_{\pm 1.07}$ |
| MTGC$^3$ | $67.39_{\pm 0.42}$ | $77.99_{\pm 0.42}$ | $66.36_{\pm 0.26}$ |

Table 2: The performance of different variants of MTGC$^3$ on different datasets.

these methods still fail to design suitable architectures for multiple tasks. Our proposed MTGC$^3$ model again outperforms all the baselines on the three datasets, demonstrating that our model is able to search for proper architectures in real-world multi-task scenarios.

## 4.3 Visualization and Analysis

To gain deeper insights of our method, we further conduct some analyses. Specifically, we generate four labels by a random MLP, GCN, GAT, and SAGE respectively in MGL, and denote the labels as $1, 3, 5, 7$. Then, we add Gaussian noises to the node representations and generate four additional labels denoted as $2, 4, 6, 8$. As a result, we generate four pairs of highly correlated tasks, i.e., label 1 and 2 are related, etc. Then, we visualize the architecture parameters, i.e., $\alpha_{ik}$, at the first layer. The results on MGL-ER are shown in Figure 3. The results show that MTGC$^3$ can search for proper architectures for different tasks. For the tasks directly generated by random GNNs (see chunks 1, 3, 5, 7), MTGC$^3$ has the highest $\alpha$ on the corresponding operation.

In addition, we visualize the absolute values of $p_{ij}$ in the soft task-collaborative module in Figure 4. Large absolute values indicate that knowledge can be transferred between tasks. For the last layer, we can observe that more knowledge can be transferred between our constructed task pairs, e.g., chunk 1 and 2, chunk 5 and 6. The results demonstrate that the soft task-collaborative module can accurately capture the transferability between tasks. Meanwhile, for the 1st layer, the absolute values exhibit more complex patterns, indicating that some complex knowledge sharing between chunks may exist. It is also worth noting that the absolute values of $p_{ij}$ at the 1st layer are generally larger than that at the last layer. This phenomenon indicates that there is more shared information at the shallow layers than at the deep layers, which is consistent with the branched sharing architecture in multi-task learning [6].

The above analyses are for the task-separated head case. We further analyze the cross-mixed head case. We visualize the architecture parameters $\alpha_{ik}$ on ToxCast dataset in Figure 5. We can observe that different chunks choose different operations, indicating that different chunks still need to find the proper operations to fit the tasks assigned through the cross-mixed head.

## 4.4 Ablation Study

In this section, we evaluate the effectiveness of each module of our framework by conducting ablation studies. We compare the following variants of our model:

- MTGC$^3$-NoStru: we use the original graph structure for all chunks.
- MTGC$^3$-Separate: we disable the collaboration between chunks, i.e., $\forall i \neq j, p_{ij} = 0$, so that different tasks search architectures separatedly.
- MTGC$^3$-FullCollab: we let $p_{ij} = 1$, i.e., full collaboration, thus the independence of each chunk is not guaranteed.
- MTGC$^3$-MLPHead: we use an MLP as the task head.
- MTGC$^3$-NoCL: we remove the task-wise curriculum training and use standard gradient descents.

We report the results of these variants on MGL-ER, Tox21, and ToxCast in Table 2. We have the following observations. Overall, our proposed full MTGC$^3$ model outperforms all the variants under all three settings, demonstrating that each component of our method is indispensable to achieve satisfactory performance in multi-task graph learning. The performance margin above MTGC$^3$-NoStru illustrates the validity of learning diverse graph structures for different chunks. Either separating all chunks by setting $p_{ij} = 0$ or allowing maximum collaboration by setting $p_{ij} = 1$ causes performance degradation, demonstrating that our soft task-collaborative module is essential to capture the transferability between tasks. In addition, if we use an MLP as the task head, the performance will drop severely, indicating the importance of the task-separated head and cross-mixed head in multi-task learning. Moreover, the task-wise curriculum training also has a consistently positive impact on the model performance.

## 5 Related Works

### 5.1 Graph neural network and multi-task learning

Message-passing GNNs [17, 39, 47, 11, 23, 22, 21, 20, 50, 51] have been proposed as an effective framework for graph machine learning following the neighborhood aggregation scheme. At each layer, nodes learn representations by aggregating their neighbors' representations. Then, the representation of the whole graph is learned by pooling all node representations [17, 47]. Graph structure learning technique [15, 44] is also applied in GNNs, which learns a better graph structure for message passing during training procedure. Graph structure learning is beneficial for enhancing the robustness of graph embedding, especially in noisy scenarios.

Multi-task learning [3, 16, 26] aims to jointly learn a set of tasks with shared parameters. Recently, some works learn to exploit similarities between tasks in GNNs to benefit multi-task graph learning. New et al. [28] explore the influence of the Hessians of each task's loss function in multi-task graph learning. SGNN-EBM [25] learns task relationships in the latent space by knowledge extracted from external datasets. MetaLink [2] uses graphs to model the relationship between data and tasks in the relational multi-task setting. Nevertheless, these works do not explore automatic architecture design in multi-task graph learning.

### 5.2 Neural architecture search

Recent years have witnessed a surge of research interest in NAS methods, which aim at designing neural architectures automatically for given tasks. Since the architecture search space is discrete, reinforcement learning (RL) [56, 29] and evolution algorithm (EA) [45**?** ] are often used in NAS methods. Besides, another strategy is transferring the discrete architecture search space into a differentiable space, e.g., DARTS [24] and SNAS [46] construct a supernet where all candidate operations are mixed to update the architecture as well as the weights simultaneously through the classical gradient descent method.

GraphNAS is gaining increasing attention from the research community [19, 7, 10, 31, 52, 33, 54, 52], including RL [7, 30], EA [49, 9], and differentiable NAS [19, 31, 32] algorithms. GraphNAS works also consider searching specific modules in graph representation learning, such as message passing layers [1], attention layers [10], and pooling operations [42]. However, existing GraphNAS methods only consider a single task and neglect the multi-task scenarios.

Outside of graphs, some multi-task NAS searches for a unified architecture for multiple tasks. However, most existing works only focus on searching the interaction operations between tasks but ignore designing different task backbones [36, 8, 37, 38, 43, 34, 48]. More importantly, all the above works focus on computer vision task but do not consider graph tasks.

# 6 Conclusion

In this paper, we propose a novel MTGC$^3$ method to tackle GraphNAS problem in multi-task scenarios. Our method searches for multiple architectures for different tasks while considering task collaboration by designing the structurally diverse supernet, the soft task-collaborative module, and the task-wise curriculum training. Extensive experiments on both synthetic and real-world datasets demonstrate that MTGC$^3$ can achieve state-of-the-art performance for multi-task graph learning. One possible limitation of this work is that our method only considers static and homogenous graphs and does not account for more complex graphs, such as dynamic or heterogenous graphs. Future research should explore methods that can effectively handle these complex graph types, in order to broaden the usage of the method in real-world applications.

## Acknowledgments and Disclosure of Funding

This work was supported by the National Key Research and Development Program of China No. 2020AAA0106300, National Natural Science Foundation of China (No. 62222209, 62250008, 62102222, 62206149), Beijing National Research Center for Information Science and Technology under Grant No. BNR2023RC01003, BNR2023TD03006, Beijing Key Lab of Networked Multimedia, China National Postdoctoral Program for Innovative Talents No. BX20220185, and China Postdoctoral Science Foundation No. 2022M711813.

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
