# Appendix for Multi-task Graph Neural Architecture Search with Task-aware Collaboration and Curriculum

**Yijian Qin**[1,2], **Xin Wang**[1,2]*, **Ziwei Zhang**[1], **Hong Chen**[1], **Wenwu Zhu**[1,2]*
[1]Department of Computer Science and Technology, Tsinghua University, [2]BNRist[†]
qinyj19@mails.tsinghua.edu.cn, {xin_wang,zwzhang}@tsinghua.edu.cn
h-chen20@mails.tsinghua.edu.cn, wwzhu@tsinghua.edu.cn

## A   Notations

Table 3: Meanings of notations.

| Notation | Meaning |
|---|---|
| $\mathcal{G}$ | The graph space |
| $\mathcal{Y}$ | The label space |
| $\mathcal{W}$ | The weight space |
| $\mathcal{A}$ | The architecture space |
| $\mathcal{O}$ | The operation search space |
| $\mathcal{L}$ | The loss function |
| $G, Y$ | A graph dataset and its corresponding labels |
| $F$ | A model mapping $\mathcal{G} \rightarrow \mathcal{Y}$ |
| $\mathbf{H}^{(l)}, \mathbf{h}_i^{(l)}$ | The node representation at the $l$-th layer (of node $i$) |
| $\mathbf{m}_i^{(l)}$ | The message aggregated to node $i$ at the $l$-th layer |
| $\mathbf{x}$ | The node representation |
| $\mathbf{A}$ | The adjacent matrix |
| $f$ | A layer in GNN $F$ |
| $o_k$ | An operation |
| $w$ | Model weight |
| $\alpha$ | The architecture parameter |
| $N$ | The number of chunks |
| $\theta$ | The trainable parameter in the soft task-collaborative module |
| $p$ | The parameter generated by Eq.(9) |
| $p'$ | The parameter generated by Eq.(11), replacing $p$ during curriculum training |
| $\delta$ | The parameter to control graph structure diversity |
| $\gamma$ | The parameter to control task-wise curriculum training |

---

*Corresponding Authors.

[†]BNRist is the abbreviation of Beijing National Research Center for Information Science and Technology.

37th Conference on Neural Information Processing Systems (NeurIPS 2023).

# B  Proofs

**Proposition 3.2.** *When using the overall loss function of multiple tasks $\mathcal{L}$ for gradient back-propagation in our framework, the partial derivative of $\mathcal{L}$ with respect to $w_{j'}$ is:*

$$\frac{\partial \mathcal{L}}{\partial w_{j'}} = \sum_i p_{ij'} \frac{\partial \mathcal{L}}{\partial f_i} \sum_{o_k \in \mathcal{O}} \alpha_{ik} \frac{\partial o_{ij'k}(\mathbf{x}_{j'}, \mathbf{A}_i)}{\partial w_{j'}}.$$

*Proof.* Here we provide the detailed derivation process of Eq.(10). Firstly we use the chain rule:

$$\frac{\partial \mathcal{L}}{\partial w_{j'}} = \sum_i \frac{\partial \mathcal{L}}{\partial f_i} \frac{\partial f_i}{\partial w_{j'}}.$$

Then we use Eq.(9) to substitute $\frac{\partial f_i}{\partial w_{j'}}$:

$$\frac{\partial \mathcal{L}}{\partial w_{j'}} = \sum_i \frac{\partial \mathcal{L}}{\partial f_i} \sum_{o_k \in \mathcal{O}} \alpha_{ik} \sum_j p_{ij} \frac{\partial o_{ijk}(\mathbf{x}_j, \mathbf{A}_i)}{\partial w_{j'}}.$$

We adjust the summation order:

$$\frac{\partial \mathcal{L}}{\partial w_{j'}} = \sum_j \sum_i p_{ij} \frac{\partial \mathcal{L}}{\partial f_i} \sum_{o_k \in \mathcal{O}} \alpha_{ik} \frac{\partial o_{ijk}(\mathbf{x}_j, \mathbf{A}_i)}{\partial w_{j'}}.$$

Since we are considering the parameters of the $j'$-th chunk at the last layer, the items with other $j$ have no contributions. We omit the items about other $j$:

$$\frac{\partial \mathcal{L}}{\partial w_{j'}} = \sum_i p_{ij'} \frac{\partial \mathcal{L}}{\partial f_i} \sum_{o_k \in \mathcal{O}} \alpha_{ik} \frac{\partial o_{ij'k}(\mathbf{x}_{j'}, \mathbf{A}_i)}{\partial w_{j'}}.$$

$\square$

# C  Experimental Settings

## C.1  Dataset Statistics

Table 4: Dataset Statistics.

| Dataset | #Graphs | Split(%) | Avg. $|V|$ | Avg. $|E|$ | #Tasks | #Classes | Metric | Type |
|---|---|---|---|---|---|---|---|---|
| MGL-ER | 1 | 50/25/25 | 5,000 | 12,500 | 8 | 5 | Accuracy | Node classification |
| MGL-WS | 1 | 50/25/25 | 5,000 | 15,000 | 8 | 5 | Accuracy | Node classification |
| MGL-BA | 1 | 50/25/25 | 5,000 | 15,000 | 8 | 5 | Accuracy | Node classification |
| OGBG-Tox21 | 7,831 | 80/10/10 | 18.6 | 19.3 | 12 | 2 | ROC-AUC | Graph classification |
| OGBG-ToxCast | 8,576 | 80/10/10 | 18.8 | 19.3 | 617 | 2 | ROC-AUC | Graph classification |
| OGBG-Sider | 1,427 | 80/10/10 | 33.6 | 35.4 | 27 | 2 | ROC-AUC | Graph classification |

## C.2  Search Space and Hyper-parameters

**Search space**. We consider a search space of standard layer-by-layer architectures without sophisticated connections such as residual or jumping connections, though our proposed method can be easily generalized. We choose five widely used message-passing GNN layers as our operation candidate set $\mathcal{O}$, including GCN [4], GAT [9], GIN [10], SAGE [2], k-GNN [5], and ARMA [3]. Besides, we also adopt MLP, which does not consider graph structures. We set the number of layers as 3 for synthetic datasets, and 5 for real-world datasets.

**Hyper-parameters**. Typically, the learning rates of $w, \alpha, \theta$ are $\eta_w = 0.0005, \eta_\alpha = 0.12, \eta_\theta = 0.05$. We set $\delta = 0.25$. In addition, $\gamma$ increases linearly from 0 to 1 during the searching procedure. The settings of hidden dimension and the number of chunks are as follows: MGL: 128 with 8 chunks; Tox21: 132 with 12 chunks; ToxCast: 128 with 16 chunks; Sider: 135 with 27 chunks. For ToxCast, we use the cross-mixed head. For the other datasets, we use the task-separate head. The hidden dimension is 339 for all baselines on all datasets.

# D  Experimental Results

## D.1  Gereral Multi-task NAS as Baselines

In this part, we compare our method with the SOTA multi-task NAS methods in recent years, inlcuding MTL-NAS [1], Sparse Sharing [7], Raychaudhuri et al. [6], AdaShare [8], and AutoMTL [11]. Since these methods are for CV tasks, we modified them as little as possible to adapt them to multi-task graph learning and our search space. The experiment results on OGBG datasets are shown in Table 5.

Table 5: The performance comparison with general multi-task NAS baselines on different datasets.

| Variant | Tox21 | ToxCast | Sider |
|---|---|---|---|
| MTL-NAS [1] | $74.77_{\pm 0.24}$ | $63.14_{\pm 0.52}$ | $55.31_{\pm 0.64}$ |
| Sparse Sharing [7] | $75.17_{\pm 1.26}$ | $64.10_{\pm 0.70}$ | $57.65_{\pm 1.15}$ |
| Raychaudhuri et al. [6] | $75.86_{\pm 0.55}$ | $62.85_{\pm 0.24}$ | $55.90_{\pm 1.25}$ |
| AdaShare [8] | $67.34_{\pm 1.08}$ | $62.91_{\pm 0.41}$ | $60.41_{\pm 0.46}$ |
| AutoMTL [11] | $73.02_{\pm 0.90}$ | $62.69_{\pm 0.39}$ | $53.94_{\pm 1.87}$ |
| MTGC[3] | $\mathbf{77.99_{\pm 0.42}}$ | $\mathbf{66.36_{\pm 0.26}}$ | $\mathbf{62.08_{\pm 1.76}}$ |

From the table, our method can outperform all the multi-task NAS baselines in the three datasets. The results demonstrate the effectiveness of our method on multi-task graph learning.

## D.2  Time Cost

**Theoretical Analysis**. Denote $|V|, |E|$ as the number of nodes and edges in the graph, respectively, and $d$ as the dimensionality of hidden representations. We denote $d_S$ as the dimensionality of the structure generation part, i.e., $\mathbf{S}$ is a matrix with the shape $d_S \times d$. The time complexity of typical message-passing GNNs in our search space is $O(|E|d + |V|d^2)$. For each chunk, calculating the graph structure by Eq.(7) costs $O(|V|d_S d + |E|d_S)$ time, and calculating all candidate operations costs $O(|\mathcal{O}|(|E|d + |V|d^2))$ time. Calculating the soft task-collaborative module costs $O(|\mathcal{O}|d^2)$ time. The overall time complexity of our method is $O(N(|\mathcal{O}|d + d_S)(|E| + |N|d))$. In practice, we use a small $d_S$ and $|\mathcal{O}|d \ge d_S$. Therefore, our model costs $O(N|\mathcal{O}|(|E|d + |V|d^2))$ time. We remark that using DARTS method in our search space has $O(|\mathcal{O}|(|E|d + |V|d^2))$ time complexity. Therefore, our method has the same time complexity as using DARTS to search for architectures for different chunks separately.

**Empirical Study**. We measure the search time of DARTS$\times N$ (using DARTS to search for architectures for different chunks separately) and our proposed method and show the results in Table 6. The two models have comparable running times, indicating our model design does not bring too much extra computational burden than DARTS$\times N$ on the searching phase. The results also confirm our complexity analysis.

Table 6: Empirical search time (NVIDIA GeForce RTX 3090).

| Dataset | Tox21 | ToxCast | Sider |
|---|---|---|---|
| MTGC[3] | 2334s | 7326s | 3201s |
| DARTS$\times N$ | 2688s | 6784s | 1917s |