# OpenReview forum: "Multi-task Graph Neural Architecture Search with Task-aware Collaboration and Curriculum"
_NeurIPS.cc/2023/Conference — NeurIPS 2023 poster_

### Official Review · Reviewer_a5eh · 2023-06-30

**Soundness:** 3 good
**Presentation:** 3 good
**Contribution:** 3 good
**Rating:** 6
**Confidence:** 5

**Summary:**

This paper puts forth an intriguing and innovative research problem: how to effectively search for graph neural architectures within multi-task domains. In addressing this question, the authors introduce MTGC, a methodology involving three primary stages. Firstly, forward propagation is carried out through the structurally diverse supernet in combination with the soft task-collaborative module. Secondly, both the architecture parameters and the soft task-collaborative module are updated. Finally, model weights are modified through task-wise curriculum learning. The aforementioned steps seem practical and are validated by their effectiveness in the experimental contexts.

**Strengths:**

(1) The work presented in this paper is well articulated and comprehensible apart from a few minor presentation issues.

(2) The paper adds a novel angle to the existing discourse by applying GNAS to a multi-task setting, which is a noteworthy approach.

(3) The research problem has been meticulously defined and the related challenges have been effectively pinpointed.

(4) The proposed methods come across as plausible. I like the manner in which the graph structure has been disentangled. It is generalizable.

(5) The experiment stage of the research is replete with ample benchmarks, which efficiently validate the effectiveness of the proposed method.

**Weaknesses:**

(1) The left part (three graphs) in Figure 2 is confusing. Are there three different input graphs? or just three disentangled graphs from the same input graph? It should be clarified.

(2) What kind of knowledge should different GNN architectures share in the multi-task setting? Can you present more discussions about this? or raise some examples?

There are some related works that should be cited.

[1] Factorizable graph convolutional networks.

[2] Automatic relation-aware graph network proliferation.

**Questions:**

As stated in weakness.

**Limitations:**

No limitation part is included.

---

> ### Author Rebuttal · Authors · 2023-08-09
>
> We thank you for your reviewing efforts and constructive comments. We address your comments point by point.
>
> *Q1: The left part (three graphs) in Figure 2 is confusing. Are there three different input graphs? or just three disentangled graphs from the same input graph? It should be clarified.*
>
> Response 1: We appreciate the reviewer's feedback regarding Figure 2. There are indeed three different input graphs in the left part of Figure 2, and each of these graphs represents the input of a task. We will make sure to provide additional clarification in the revised version to avoid any ambiguity.
>
> *Q2: What kind of knowledge should different GNN architectures share in the multi-task setting? Can you present more discussions about this? or raise some examples?*
>
> Response 2: We appreciate the reviewer's value question about the shared knowledge in our multi-task architecture. If the labels of two tasks have a high correlation, they will share more information inside. For example, if the hidden feature of a chunk's output in a layer is useful for the inference of two different tasks, the supervised signals of both tasks will further optimize the learning of the parameters of this chunk. Although the form of the shared knowledge is parameter that cannot be intuitively understood. The strength of the sharing can be estimated by $p_{ij}$ in our framework, as shown in Figure 4.
>
> *Q3: There are some related works that should be cited.*
>
> Response 3: Thank you for your suggestion, we will ensure that the revised manuscript includes the references of these papers to address the reviewer's concern.

---

### Official Review · Reviewer_cm5U · 2023-07-04

**Soundness:** 2 fair
**Presentation:** 2 fair
**Contribution:** 2 fair
**Rating:** 4
**Confidence:** 4

**Summary:**

This paper proposes a method called MTGC3 for searching GNNs in multi-task learning. Firstly, it highlights the importance of designing different GNNs for different tasks. Then, it introduces the Structurally Diverse Supernet and Soft Task-Collaborative modules, which enable the generation of task-specific architectures that can operate separately or collaboratively. The task-wise training strategy is employed to address the task-imbalance problem.


**Strengths:**

The paper presents an interesting and novel approach that jointly searches for different architectures for different tasks, considering task-specific and shared information.


**Weaknesses:**

Method design:

The relationship with GNNs is not clear, and further comparisons with general multi-task+NAS methods should be discussed.
 While this paper seems to be the first to propose a method for searching multi-task GNNs, the designed method (i.e., separated trunk, soft collaboration module, and task-wise learning) appears to have weak connections with GNNs. Why can't existing NAS+MTL methods be applied? From this perspective, the first contribution of this paper appears to be weakened.

In terms of experiments, the evaluations of the proposed method could be further improved:

1. Performance comparisons with general multi-task methods should be included, such as in Line 316.
2. In Table 2, the results of the ablation study are remarkably similar to each other, indicating that they represent the basic functionality of the designed modules in this paper.
3. Evaluation of the Cross-mix head. This module is proposed to allocate sufficient hidden units for each task. When evaluating this module, it seems necessary to remove the masked tensors and utilize only a few units for each task instead of showing the α_{ik} values?

In summary, the key contribution of this paper is not adequately justified. The paper overlooks MTL+GNN baselines, which are essential for demonstrating the effectiveness of the proposed method.

**Questions:**

Please check the weakness.

**Limitations:**

Please check the weakness.

---

> ### Author Rebuttal · Authors · 2023-08-10
>
> We thank you for your reviewing efforts and constructive comments. We address your comments point by point.
>
> *Q1: The relationship with GNNs.*
>
> Response 1: Thank you for the comment. Here we would like to furtherly clarify that we designed our method based on the specific requirements of graph multi-task learning scenarios.
>
> (1) Our approach considers different graph structures for different tasks, which is a critical consideration in graph multi-task settings. Our structurally diverse supernet enables different tasks to learn with different graph structures, which is not considered in existing general multi-task NAS methods that mainly focus on searching for sharing parts.
>
> (2) Our approach contains practical considerations for graph scenarios, such as backbone search and handling a large number of tasks. The semantic information of the multi-tasks in the graph domain is diverse and complex, leading to significant performance differences across GNNs. Figure 1 in our paper demonstrates these disparities. Existing general multi-task NAS methods[1][2][3][4][5] mainly focus on searching for sharing parts while manually fixing the backbone. In contrast, our method combines backbone searching and shared parameter searching to effectively address the unique challenges of graph multi-task problems. We also incorporate a cross-mixed head design that significantly reduces the number of parameters required for scenarios with an extremely large number of tasks, which is a practical consideration that is not addressed in existing general multi-task NAS methods.
>
> In the revised version, we will include the discussion that addresses the unique challenges in graph multi-tasks NAS, which motivates the need for a specialized approach. We also acknowledge that our method can be extended to other domains by considering more domain-specific priors. By providing this analysis, we aim to strengthen the justification for our proposed approach and clarify the distinct contributions of our work.
>
> *Q2: Performance comparisons with general multi-task methods.*
>
> Response 2: We appreciate the reviewer's comment. Besides, we have compared our method with several representative general multi-task methods such as MTL-NAS[1], Sparse Sharing[2], Raychaudhuri et al.[3], AdaShare[4] and AutoMTL[5]. The results are shown in Appendix D due to the space limit.
>
> | Method | Tox21 | ToxCast | Sider |
> | --- | --- | --- | --- |
> | MTL-NAS | $74.77_{0.24}$ | $63.14_{0.52}$ | $55.31_{0.64}$ |
> | AdaShare | $67.34_{1.08}$ | $62.91_{0.41}$ | $60.41_{0.46}$ |
> | AutoMTL | $73.02_{0.90}$ | $62.69_{0.39}$ | $53.94_{1.87}$ |
> | Sparse Sharing | $75.17_{1.26}$ | $64.10_{0.70}$ | $57.65_{1.15}$ |
> | Raychaudhuri et al.[5] | $75.86_{0.55}$ | $62.85_{0.24}$ | $55.90_{1.25}$ |
> | MTGC3 | $78.01_{0.68}$ | $66.74_{0.57}$ | $62.26_{1.42}$ |
>
> The results demonstrate the effectiveness of our methods on multi-task graph learning. We will revise this section in the revision.
>
> [1] MTL-NAS: Task-Agnostic Neural Architecture Search towards General-Purpose Multi-Task Learning. CVPR 2020.
>
> [2] Learning Sparse Sharing Architectures for Multiple Tasks. AAAI 2020.
>
> [3] Controllable Dynamic Multi-Task Architectures. CVPR 2022.
>
> [4] Adashare: Learning what to share for efficient deep multi-task learning. NIPS 2020.
>
> [5] AutoMTL: A Programming Framework for Automating Efficient Multi-Task Learning. NIPS 2022.
>
> *Q3: The results of the ablation study are remarkably similar to each other, indicating that they represent the basic functionality of the designed modules in this paper.*
>
> Response 3: We appreciate the reviewer's comments. In order to provide a clearer illustration of the contributions of the modules. We conducted another experiment with the variant model MTGC^3-NoAll (NoStru+FullCollab+MLPHead+NoCL). This variant model only keeps the separate chunks with different architectures in our proposed design. The results are as follows:
>
> | Method | MGL-WS | Tox21 | ToxCast |
> | --- | --- | --- | --- |
> | DARTS| $64.16_{0.29}$ | $76.96_{0.57}$ | $65.23_{0.60}$ |
> | MTGC^3-NoAll | $66.68_{0.32}$ | $77.39_{0.20}$ | $65.62_{0.61}$ |
> | MTGC^3-NoStru | $67.17_{0.60}$ | $77.42_{1.00}$ | $66.30_{0.30}$ |
> | MTGC^3-FullCollab | $67.21_{0.55}$ | $77.83_{0.71}$ | $66.01_{0.50}$ |
> | MTGC^3-MLPHead | $64.47_{1.16}$ | $77.62_{0.57}$ | $64.61_{0.53}$ |
> | MTGC^3-NoCL | $67.15_{0.52}$ | $77.68_{0.48}$ | $66.00_{1.07}$ |
> | MTGC3 | $67.39_{0.42}$ | $77.99_{0.42}$ | $66.36_{0.26}$ |
>
> Compared with DARTS, we find this variant does achieve a stable improvement in performance, indicating that the proposed key idea (using different architectures in different chunks for different tasks) contributes a lot. All other designs in our paper are based on this key idea. And the performance can still be improved by these designed modules. Although the improvements they bring may not be as significant as the main contributing factor, they are still valuable for the overall framework. Besides, we find that in some cases MTGC^3-MLPHead behaves even worse than MTGC^3-NoAll, this may be due to that our designed modules are led to poorer learning when receiving more mixed gradients with the MLP classification head. We will add the experiments and discussions in the revised version.
>
> *Q4: Evaluation of the Cross-mix head.*
>
> Response 4: We appreciate the reviewer's suggestion regarding the evaluation of the Cross-mix head module and the allocation of hidden units for each task. In our implementation, we use 128 dimensions for ToxCast and $p=1/16$ in the Bernoulli distribution. Therefore, 8 units are assigned to each task in expectation. Following your suggestion, we conduct experiments with a few units for each task. We pre-assigned only 2 units for each task. The evaluation metric on ToxCast is $61.70_{0.58}$, while that of our method is $66.36_{0.26}$, indicating that only using very few units for each task does not work well in this case. Our cross-mixed head is a reasonable solution to this situation.

---

### Official Review · Reviewer_Eks4 · 2023-07-07

**Soundness:** 3 good
**Presentation:** 3 good
**Contribution:** 3 good
**Rating:** 7
**Confidence:** 4

**Summary:**

Existing GraphNAS algorithms search for well-performing architectures for a single task, this paper searches architectures for multiple graph task at the same time to share common knowledges. Specifically, this paper use structurally diverse supernet and soft task-collaborative module to search for the optimal architectures and the collaborative pattern of different tasks. The paper further proposes to leverage curriculum learning to balance gradient scales of different tasks during searching. Empirical results show that this method can improve GraphNAS in multi-task scenarios.


**Strengths:**

1. Different from existing GraphNAS papers, this paper introduces a complete framework to search for multiple architectures for multi-tasks at the same time.
2. This paper introduces curriculum learning on multi-task scene to balance task gradients in searching phase.
3. The experiments on different datasets and ablation study are sufficient to show how the designed method works.



**Weaknesses:**

1. Some designs of this paper are not well supported. At the beginning of Section 3 the authors present Assumption 1, which is the basis of the structurally diverse supernet. The authors also verified their assumption through experiments that different tasks require different architectures. However, the graph structure differences for different tasks are proposed in the supernet and are not experimentally verified.
2. In Section 4.3, the absolute value of p is shown in Figure 4. Do positive and negative values have the same meaning of transferred knowledge? Please give more explanation of the meaning.
3. In the algorithm proposed by the authors there are many hyperparameters that need to be optimized, such as the learning rates of different parts. This may lead to difficulties in tuning when switching between different datasets.


**Questions:**

See weakness.

**Limitations:**

No.

---

> ### Author Rebuttal · Authors · 2023-08-10
>
> We thank you for your reviewing efforts and constructive comments. We address your concerns point by point.
>
> *Q1: The graph structure differences for different tasks are proposed in the supernet and are not experimentally verified.*
>
> Response 1: Thank you for the comment. Following your suggestion, we validate the necessity of graph structure differences for different tasks as we did for architecture differences. To maintain consistency with the process in our model, we first randomly sampled 12 pairs of different $S_u, S_v$. We train all these tasks separated with the best architecture searched by DARTS algorithm and these pairs of $S_u, S_v$. These $S_u, S_v$ are fixed during training, and we use them to generate edge weights as in Equation (7). We rank these pairs according to their performance and calculate the Kendall rank correlation between rankings of different tasks as well. We find the Kendall correlation values are very low. In Tox21, 73.6% values are less than 0.2, 47.2% values are less than 0. In ToxCast, 78.6% values are less than 0.2, and 47.0% values are less than 0. The results indicate that different graph structures also behave differently on different tasks, illustrating the necessity of graph structure differences. We will add the experiment and results in the revised version.
>
> *Q2: In Section 4.3, the absolute value of p is shown in Figure 4. Do positive and negative values have the same meaning of transferred knowledge? Please give more explanation of the meaning.*
>
> Response 2: We appreciate the reviewer's comment regarding the interpretation of positive and negative values in Figure 4. We can consider $p_{ij}$ as a part inside the operation $o_{ijk}$. Since the parameters in $o_{ijk}$ can be negative, $p_{ij}$ can also be negative. Positive or negative $p_{ij}$ represents the embedding is positively or negatively correlated with downstream parameters While positive and negative values of $p_{ij}$ represent different magnitudes, they both indicate the presence of transferred knowledge. The absolute value is used to emphasize the strength or magnitude of the transferred information, regardless of its direction (positive or negative). The sign of $p_{ij}$ only represents the direction of it. We will provide a detailed explanation in the revised version to clarify the meaning of the absolute value of $p_{ij}$.
>
> *Q3: Difficulties in tuning when switching between different datasets.*
>
> Response 3: Thank you for the comment. We have provided the typical hyper-parameter settings, including the learning rates of different parts, in Appendix C. Besides, we explore the sensitivity of these hyper-parameters. The results are shown below.
>
> | $\eta_S$ | 0.0008 | 0.001 | 0.0012 |
> | --- | --- | --- | --- |
> | MGL-WS | $67.01_{0.66}$ | $67.39_{0.42}$ | $67.13_{0.48}$ |
>
> | $\eta_w$ | 0.004 | 0.005 | 0.006 |
> | --- | --- | --- | --- |
> | MGL-WS | $66.74_{0.52}$ | $67.39_{0.42}$ | $67.40_{0.63}$ |
>
> | $\eta_\alpha$ | 0.01 | 0.012 | 0.014 |
> | --- | --- | --- | --- |
> | MGL-WS | $67.54_{0.52}$ | $67.39_{0.42}$ | $66.83_{0.47}$ |
>
> The results demonstrate that our method is not very sensitive to these hyper-parameters, indicating that they can be easily tuned.

---

> > ### Comment · Reviewer_Eks4 · 2023-08-18
> >
> > I appreciate the authors for their comprehensive response. The additional experiments provided by the authors on the need for graph structure differences for different tasks and hyperparameters resolve my concerns. I think this is overall a good work and will be happy to increase my score to 7.

---

> > > ### Author Response · Authors · 2023-08-18
> > > **Thanks for the follow-up**
> > >
> > > We thank the reviewer for the detailed check and response to our rebuttal content, and we believe this fruitful rebuttal further improves our paper.

---

### Official Review · Reviewer_sAxj · 2023-07-08

**Soundness:** 3 good
**Presentation:** 3 good
**Contribution:** 3 good
**Rating:** 7
**Confidence:** 4

**Summary:**

This paper proposes a graph multi-task neural architecture search technique, which is a new scene in graph NAS. This paper takes some reasonable measures to handle the problem, including structurally diverse supernet, soft task-collaborative module, and task-wise curriculum training. Its performance is worthy of recognition.

**Strengths:**

- The proposed supernet and collaborative module of this article are well-motivated.

- This article introduces task-wise curriculum learning, which make sense in the multi-task NAS problem.

- The experimental results in this paper are very good, which show the mechanism of the method clearly.

- This article performs a full ablation analysis.

**Weaknesses:**

- Multi-task NAS is not a new technique in other NAS area. Prior arts have had some exploration. It is better to provide more comparison with those methods, especially the issue of how to exchange information between different tasks.

- In the article, the definition of search space is vague. Please give more details of it.

- Figure 2 is pleasing but hard to understand. It proves to be rather challenging to comprehend the inner workings of the framework and what the input-output formats are for each module.

**Questions:**

Please see the Weaknesses.

**Limitations:**

None.

---

> ### Author Rebuttal · Authors · 2023-08-09
>
> We thank you for your reviewing efforts and constructive comments. We address your comments point by point.
>
> *Q1: Multi-task NAS is not a new technique in other NAS area. Prior arts have had some exploration. It is better to provide more comparison with those methods, especially the issue of how to exchange information between different tasks.*
>
> Response 1: Thank you for the comment. Multi-NAS has been indeed explored in areas like CV and NLP. However, we designed our method based on the specific requirements of graph multi-task learning scenarios.
>
> (1) Our approach considers different graph structures for different tasks, which is a critical consideration in graph multi-task settings. Our structurally diverse supernet enables different tasks to learn with different graph structures, which is not considered in existing general multi-task NAS methods that mainly focus on searching for sharing parts.
>
> (2) Our approach contains practical considerations for graph scenarios, such as backbone search and handling a large number of tasks. The semantic information of the multi-tasks in the graph domain is diverse and complex, leading to significant performance differences across GNNs. Figure 1 in our paper demonstrates these disparities. Existing general multi-task NAS methods[1][2][3][4][5] mainly focus on searching for sharing parts while manually fixing the backbone. In contrast, our method combines backbone searching and shared parameter searching to effectively address the unique challenges of graph multi-task problems. We also incorporate a cross-mixed head design that significantly reduces the number of parameters required for scenarios with an extremely large number of tasks, which is a practical consideration that is not addressed in existing general multi-task NAS methods.
>
> (3) Our design of information exchange between different tasks is new. Our proposed soft task-collaborative module captures the complex relationships between tasks and is specifically designed for the supernet, allowing for simultaneous optimization. Furthermore, our task-wise curriculum training strategy is tailored to our layer-wise disentangle network, and our re-weighing technique rebalances the partial derivatives from different tasks within our framework. Extensive experiments further support the effectiveness and superiority of our method.
>
> Besides, we have compared our method with several representative general multi-task methods such as MTL-NAS[1], Sparse Sharing[2], Raychaudhuri et al.[3], AdaShare[4] and AutoMTL[5]. The results are shown in Appendix D due to the space limit. The results demonstrate the effectiveness of our methods on multi-task graph learning.
>
> In the revised version, we will include the discussion that addresses the unique challenges in graph multi-tasks NAS, which motivates the need for a specialized approach. We also acknowledge that our method can be extended to other domains by considering more domain-specific priors. By providing this analysis, we aim to strengthen the justification for our proposed approach and clarify the distinct contributions of our work.
>
> [1] MTL-NAS: Task-Agnostic Neural Architecture Search towards General-Purpose Multi-Task Learning. CVPR 2020.
>
> [2] Learning Sparse Sharing Architectures for Multiple Tasks. AAAI 2020.
>
> [3] Controllable Dynamic Multi-Task Architectures. CVPR 2022.
>
> [4] Adashare: Learning what to share for efficient deep multi-task learning. NIPS 2020.
>
> [5] AutoMTL: A Programming Framework for Automating Efficient Multi-Task Learning. NIPS 2022.
>
> *Q2: In the article, the definition of search space is vague. Please give more details of it.*
>
> Response 2: Thank you for the comments, we introduce the candidate operators in Section 2.2, we contain GCN, GAT, GIN, SAGE, k-GNN, ARMA, and MLP in our search space. The entire GNN backbone is a layer-by-layer architecture without sophisticated connections.
>
> *Q3: Figure 2 is pleasing but hard to understand. It proves to be rather challenging to comprehend the inner workings of the framework and what the input-output formats are for each module.*
>
> Response 3: We appreciate the reviewer's feedback regarding Figure 2. The structurally diverse supernet is the backbone of the framework, the input is the graphs of different tasks, and the output is the prediction for the target tasks. The soft-collaborative module is a part that contains learnable parameters in the supernet. The task-wise curriculum learning strategy is the optimization method that controls the calculation of gradients. We will ensure that the revised manuscript includes an improved Figure 2 with accompanying descriptions for better clarity and understanding.

---

> > ### Comment · Reviewer_sAxj · 2023-08-18
> > **Raise my score**
> >
> > Thank you for your thorough rebuttal and addressing my concerns. I have carefully considered your responses and revised my assessment of the paper.
> >
> > The paper initially lacked sufficient comparison with prior arts in the multi-task NAS field. However, the authors have addressed this concern by providing comparisons with existing general multi-task NAS methods in Appendix D and clarifying the differences between the approach of this work and the general Multi-task NAS approach, including aspects of model design and applicable scenarios. I think highlighting this part of the main paper could have emphasized the value of this paper even more. Other concerns are also addressed in the rebuttal.
> >
> > Overall, I appreciate the authors' efforts in addressing my comments and improving the paper. I'd like to raise my score to show my support.

---

> > > ### Author Response · Authors · 2023-08-18
> > > **Thanks for the follow-up**
> > >
> > > Thank you for the suggestions and response to our work and the rebuttal content. We believe with the rebuttal content, our paper will be made more clear.

---

### Official Review · Reviewer_xv8u · 2023-07-08

**Soundness:** 3 good
**Presentation:** 3 good
**Contribution:** 3 good
**Rating:** 8
**Confidence:** 5

**Summary:**

This paper proposes a multi-task graph NAS approach by learning the relationships between tasks. It uses the structurally diverse supernet to learn multiple architectures and structures together, the soft task-collaborative module to learn task relationships to exchange information, and then use task-wise curriculum training to balance task difficulties. Empirical results on OGB datasets are strong.

**Strengths:**

1. This paper is well-organized and easy to follow.

2. This paper introduces the multi-task graph NAS problem, which is important in graph learning, yet unexplored in previous works.

3. The motivations of the proposed three designs are good. They are reasonable for solving the problem.

4. The results on OGB datasets are good compare with SOTA baselines.

**Weaknesses:**

1. After calculating the edge weights, how do you perform the architectures using these edge weights in the continuous space? The implementation of this part is missing in the paper.

2. Lack of detailed motivation for some parts of the methods. In Equation 9, there are other functions that can also be chosen to represent the relationship between tasks such as sigmoid, why tanh function is used here?

3. For the soft task-collaborative module, the parameters $\theta$ are learned in a continuous space. How do you keep them in the final architecture?

**Questions:**

Check the weaknesses parts.

---

> ### Author Rebuttal · Authors · 2023-08-09
>
> We thank you for your reviewing efforts and constructive comments. We address your comments point by point.
>
> *Q1: After calculating the edge weights, how do you perform the architectures using these edge weights in the continuous space? The implementation of this part is missing in the paper.*
>
> Response 1: We appreciate the reviewer's comments. The edge weights represent the importance of different weights. Once the edge weights are calculated, we multiply the weights on the message passed by this edge. If we denote $w_{ij}$ as the edge weights between node $i$ and $j$, then the message passing process shown in Equation (1) can be changed to:
>
> $$\mathbf{m_i}^{(l)} = \text{Agg}(w_{ij}\mathbf{h}_j^{(l)}|j\in \mathcal{N}_i)$$
>
> In this way, we can use edge weights for all GNN operations. We will add a detailed explanation of how to use the edge weights in the revised version.
>
> *Q2: Lack of detailed motivation for some parts of the methods. In Equation 9, there are other functions that can also be chosen to represent the relationship between tasks such as sigmoid, why $\tanh$ function is used here?*
>
> Response 2: We appreciate the reviewer's feedback. In Equation 9, the choice of the $\tanh$ function is motivated by its desirable properties for capturing task relationships within our framework. We have the reasons as follows:
>
> (1)	Symmetry around the origin: The $\tanh$ function is symmetric around the origin, which allows it to model both positive and negative relationships between tasks. This is particularly important as tasks can exhibit different types of relationships, including positive correlations, negative correlations, or no correlations at all.
>
> (2)	Bounded output range: The $\tanh$ function outputs values between -1 and 1, which provides a bounded range for representing the strength of task relationships. This range can be interpreted as the degree of collaboration or interdependence between tasks, with values closer to 0 indicating weaker relationships and values closer to -1 or 1 indicating stronger relationships.
>
> In summary, $\tanh$ function has great properties which exactly match the demand of task relationship representation in our framework.
>
> *Q3: For the soft task-collaborative module, the parameters $\theta$ are learned in a continuous space. How do you keep them in the final architecture?*
>
> Response 3: We appreciate the reviewer's question. In our soft task-collaborative module, the parameters $\theta$ are indeed learned in a continuous space. After the whole training procedure, we keep the continuous values of $\theta$ in the final architecture, and we evaluate the performance directly with the parameters in the supernet.

---

> > ### Comment · Reviewer_xv8u · 2023-08-19
> > **Thanks for you reply.**
> >
> > My concerns have been resolved, thanks. It is an interesting paper, and I would like to increase my score.

---

> > > ### Author Response · Authors · 2023-08-19
> > > **Thanks for the follow-up**
> > >
> > > Thank you very much for your valuable comments and the careful check of our rebuttal. We believe this discussion will greatly contribute to our paper.

---

### Decision · Program_Chairs · 2023-09-21

**Decision:**

Accept (poster)

**Comment:**

This paper proposes a new multi-task graph neural architecture search method.

Four reviewers gave acceptance scores while satisfied with its task definition, method design, and thorough experimental results.
On the contrary, one reviewer raised the insufficient evidence of the proposed contributions.

During the rebuttal, the authors provided additional results on the concerns, and AC judged that the authors successfully addressed the issues.

Considering the scores and the author response, AC recommends accepting this paper.